# Bubble-Proof Algorithm for Wave Union TDCs

Paweł Kwiatkowski *, Dominik Sondej and Ryszard Szplet

Faculty of Electronics, Military University of Technology, 00-908 Warsaw, Poland;
dominik.sondej@wat.edu.pl (D.S.); ryszard.szplet@wat.edu.pl (R.S.)
*   Correspondence: pawel.kwiatkowski@wat.edu.pl

**Abstract:** Nowadays state-of-the-art time-to-digital converters (TDCs) are commonly implemented in field-programmable gate array (FPGA) devices using different variations of the wave union method. To take full advantage of this method many design challenges need to be overcome, one of which is an efficient data encoding. In this work, we describe in detail an effective algorithm to decode raw output data from a newly designed multisampling wave union TDC. The algorithm is able to correct bubble errors and detect any number of transitions, which occur in the wave union TDC output code. This allows us to reach a mean resolution as high as 0.39 ps and a single shot precision of 2.33 ps in the Xilinx Kintex-7 FPGA chip. The presented algorithm can be used for any kind of wave union TDCs and is intended for partial hardware implementation.

**Keywords:** field programmable gate arrays; time measurements; time-to-digital converter (TDC); wave union TDC; wave union encoder





## 1. Introduction

Wave union is one of the most popular methods for implementation of high-resolution time-to-digital converters (TDCs) in field programmable gate array (FPGA) devices. This method was first proposed by Wu and Shi in 2008 [1], and since then it has been used for various applications including nuclear physics [2,3], time interval counters [4], light detection and ranging (LiDAR) systems [5]. It has also undergone significant improvement during this time, making it possible to obtain the highest performance of TDCs in the FPGA technology in terms of the measurement rate, time resolution and precision [6–8]. However, in order for this to be possible, several design challenges have to be overcome. An efficient encoding scheme is one of them [6,9].

The idea of the wave union method is to feed a pulse train (a wave union, hence the name of the method) to the input of a tapped delay line (TDL) whenever the start hit occurs. Then the stop signal latches the current logic states of the delay line taps in a related register. The start and stop pulses represent the beginning and end of the measured time interval, respectively. As a result of the conversion, the register contains alternating sequences of high and low logic states. Information about the duration of the measured time interval is included in bit positions where these states change (0-1 and 1-0 transitions). This process is further hindered by occurrence of the so-called bubble errors. The bubble error is defined as a missing logic state "1" (or "0") around the expected transition and is a common phenomenon in flash analog-to-digital converters (ADC) [10]. Currently, there can be observed increasing requirements for TDC converters in terms of resolution, number of channels and measurement rate. These needs significantly increase the amount of measurement data and therefore an encoder hardware implementation is desirable.

In [9] Wu proposed a 3-input AND gate-based encoder that detects pattern "001" instead of "01" to solve single bubble errors and convert non-thermometer code to one-out-of-N code. A similar approach, a folded thermo-to-binary encoder with NAND gate-based bubble suppression, was used in [11]. A stepped-up tree encoder can be used in a second stage to get the output data in natural binary code [12]. A ones-counter encoding scheme was used to effectively decode data from multi-chain merged TDLs [13]. The same idea

was then adopted into the wave union TDC [14]. All of these designs are in fact similar to the solutions already used in flash ADCs [10]. With regard to the wave union TDCs, the mentioned solutions allow to identify only a limited number of transitions (typically 1 or 2). Moreover, some implementations in very high process node FPGA devices (e.g., 20 nm CMOS) consider only two transitions [15,16]. However, the highest precision converters often use multiple transitions, e.g., 6 [4] or 8 [8]. A certain solution to this expectation is the pre-encoder presented in [12]. The encoder is divided into so-called clusters and each of them is capable to detect 1-0 transitions. The proposed solution is intended to detect only one transition in a complete code but this idea (clustering) is extendable to more transitions [8]. A similar approach, multiple six taps windows (clusters) on the thermometer code value, was proposed in [17]. Clusters can be also used to solve simple bubble errors (e.g., single and double bubbles as in [4]). However, using most advanced FPGA devices for TDC implementation it is common to observe bubbles even on eight subsequent bits [18] and then this method becomes insufficient. The clustering method can be combined with decomposition [18] (simultaneously proposed by another research group and called a sub-TDL [19]) to correct bubble errors. It is worth mentioning that all of these works investigate identification of a determined number of transitions. Even when more transitions occur, as is often the case for the wave union (WU) TDC with the infinite step response (a wave union launcher type B [1]), some data (transitions) are lost to facilitate data encoding [1,12]. As the method applied to encode the result from the TDC affects the final performance of the conversion, it has to be carefully selected [20].

Recently, we have presented a new TDC architecture that combines the wave union with multisampling [21]. This converter was capable of reaching subpicosecond resolution (0.69 ps) but we have encountered significant problems with efficient data encoding. We have further improved the TDC design and, in this work, we present a new algorithm used to calculate values of time intervals based on the raw TDC output data. The new encoding scheme deals effectively with a different number of transitions that can occur in raw data, which has not yet been investigated in previous papers. The presented algorithm can be used to efficiently encode raw output data of any type of the WU TDCs, including converters with finite and infinite step responses (wave union type A and B, respectively) [1], as well as with multiedge coding in independent coding lines [4] (also known as the super wave union [16]).

## 2. Multisampling Wave Union TDC

Figure 1 presents a block diagram of a multisampling wave union (MSWU) TDC. The circuit consists of a TDL, four registers, and two wave union launchers. Comparing to [21], in this work two different types of launchers for start and stop signals were used. In the start signal path we have used a type B wave union launcher (with the infinite step response). This means that the start pulse initializes continuous generation of pulses that are fed to the tapped delay line. The stop signal is applied to a type A wave union launcher (with the finite step response). This launcher always creates 2 pulses (four edges: two rising and two falling), which are used to latch four TDL samples in the registers. As each edge of these pulses is shifted in time then the registers contain four different snapshots of a wave union that propagates along the TDL. The first snapshot is latched in Register 1 and then passed to Register 3 on the second rising edge of the wave union in the stop signal path. Similarly, a second snapshot is latched in Register 2 and then passed to Register 4. Both wave union launchers were built using look-up tables (LUTs) configured as logic gates presented in Figure 1. The first launcher (WU launcher B) is a NAND gate-based startable oscillator while the second one (WU launcher A) contains two edge detectors. Proper timing was ensured by the use of manual placing and routing of these elements in FPGA. The use of an additional launcher in the stop signal path together with a set of registers allows increasing the amount of information about the measured time interval, and thus improving measurement resolution and precision. This is in contrast to super WU TDCs [4,16] where performance increase is obtained at the expense of an intensive logic

resource usage. Continuous generation of pulses in the start path leads to a significant complication of data encoding. So far this problem has been omitted by taking into account only one or two transitions. Here we will show how to detect all transitions to improve TDC performance.

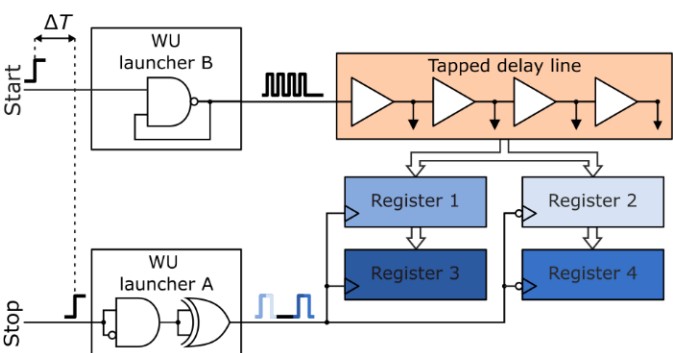

**Figure 1.** Multisampling wave union TDC block diagram.

### 3. Test Setup

Two MSWU TDCs were implemented in a Xilinx Kintex-7 FPGA chip mounted in an MTC 108 multichannel time interval counter [22]. Each TDL consists of 200 carry chain multiplexers that ensure the fastest connection paths between subsequent elements in programmable logic blocks. The TDL is connected to four registers made of 200 flip-flops each. The MTC 108 counter contains, i.a., a frequency synthesizer configured to multiply the 10 MHz reference clock signal to frequency of 700 MHz. Then, a 700 MHz clock is used in the FPGA device as a stop signal for the TDCs. Each TDC measures a time interval between the occurrences of hits, related to rising edges of measured signals, and the nearest rising edges of the high frequency clock signal. To test newly developed algorithms, described in detail in the following sections, we have prepared a test setup presented in Figure 2.

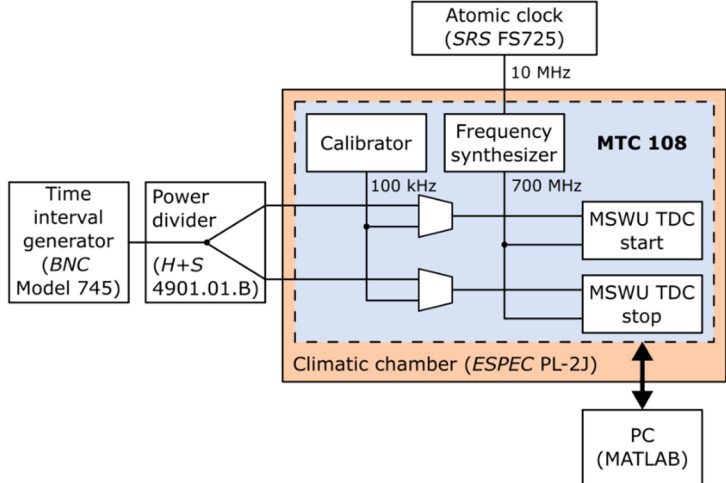

**Figure 2.** Test setup.

The MTC 108 counter was placed inside a PL-2J climatic chamber from ESPEC. The chamber provided both a stable temperature (21 °C) for a long time and additional protection against potential external electromagnetic interferences. As the reference clock we have used the rubidium standard clock FS725 (Stanford Research Systems) that provides a highly stable 10 MHz signal (one second Allan Variance not worse than $2 \times 10^{-11}$). A single signal from the Model 745 time interval generator (Berkeley Nucleonics) was split into two counter channels using the 4901.01.B power divider (Huber+Suhner). During the

calibration procedure, the counter performed the statistical code density test (SCDT) [23] using 100,000 samples from a calibrator circuit. The calibrator is an adjustable square wave generator that produces an unstable signal with a frequency of around 100 kHz. This signal is asynchronous with regard to the 700 MHz base clock and meets the requirements for the SCDT presented in [24]. Raw calibration data ($4 \times 200$ bits for each hit and each TDC) were sent via the USB interface to the control computer. Dedicated scripts in MATLAB have been developed that process data using two pre-encoders and one common post-encoder in order to: (1) accurately identify current transfer functions of the TDCs using a calibration signal and then, (2) calculate the values of measured time intervals, generated by an external source (BNC Model 745 and the splitter).

When splitting a single signal into two channels, the jitter of the reference time interval is virtually removed because the same signal edge is registered in both TDCs. In fact, the TDCs measure an offset between two channels in this case (cable lengths are approximately the same). Assuming that this offset is relatively small, also the error due to limited stability of the clock signal is minimized. Thus, the presented test setup can be regarded as the best scenario where possibly the highest precision can be achieved. Additionally, we have also tested MSWU performance by measuring different time intervals generated with the aid of the BNC Model 745 delay generator.

## 4. Encoding Algorithms

The encoding problem related to the wave union method is presented in Figure 3. For simplification, only 40 bits are investigated ($i_{max} = 40$) instead of all 200. Let us assume that two pulses (two rising and two falling edges) propagate along the TDL (Figure 3a). Then the raw data ($D$) latched in the register can look like a string of zeros and ones presented in Figure 3b. Logic states marked in red indicate bubble errors, i.e., unexpected logic state "1" or "0", which occurs shortly after the signal's transitions. In [18] Song et al. proposed a parameter called max bubble depth (MBD) to indicate the maximum number of bits near the expected transition that can be susceptible to bubble errors. The value of the MBD depends on a particular design: FPGA technology used, environmental conditions (process, voltage and temperature—PVT), implementation details (placement and routing), etc. In our case, we have found it experimentally to be equal to 5 (as it is also presented in Figure 3). Binary data presented in Figure 3b should contain two transitions 0-1 and two transitions 1-0, so a maximum number of transitions ($t_{max}$) is 4. Figure 3c presents the data after bubble suppression. The bits with bubble errors were reordered (marked in grey and underlined), which enables correct identification of the transition positions ($R(t)$, where $t$ indicates transition number). Each $R(t)$ can be treated as a single measurement, but combined with others improves TDC performance.

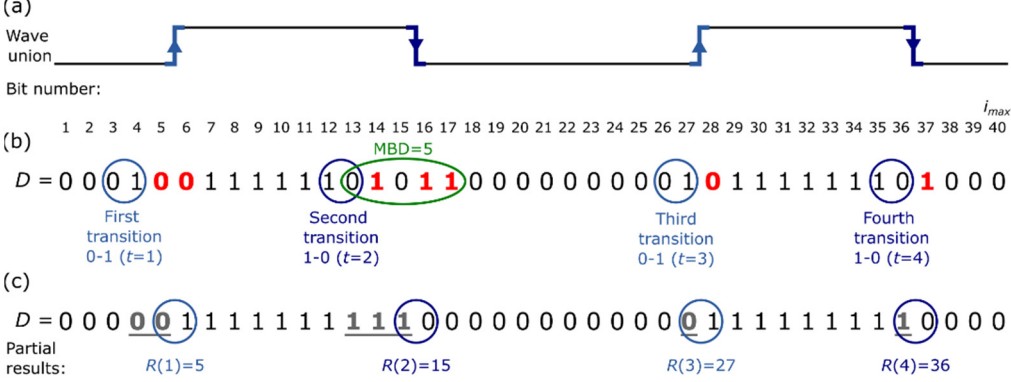

**Figure 3.** Data encoding problem: (**a**) wave union shape inside the TDL, (**b**) raw data latched in a register (bubble errors marked red), (**c**) data after bubble suppression (reordered bubbles marked grey and underlined).

The use of the type B wave union launcher in the start path causes that depending on the time interval duration the first register and the second one can contain a different number of transitions (in the designed TDC from 4 for the shortest time intervals up to 8 for time intervals close to the TDC operation range, i.e., 1.43 ns). We have prepared two 2-stage algorithms in MATLAB software to find optimal encoding scheme that can be easily implemented in hardware. First, we need to identify each transition position ($R(t)$) in all four registers. Then, it is necessary to combine data from all registers into one result. For the first stage, i.e., for a pre-encoder, we have taken two approaches: (1) counter-based and (2) decomposition with clustering-based. The first approach is more straightforward to analyze and easier to implement in software but is not easily transferable to hardware. The second approach is preferred for hardware implementation but to be sure that it is properly developed it has been compared with the counter-based encoder. The second stage of the algorithm, i.e., a post-encoder that combines all $R(t)$ from all registers, is common for both pre-encoders.

### 4.1. Pre-Encoder: Counter-Based

Operation principle of the first pre-encoder is presented in Figure 4. To simplify, the pre-encoder adds the number of logic states "1" or "0" in subranges whose boundaries are determined by the first identified transition position extended by the MBD value. In detail, the pre-encoder checks bit-by-bit ($i$ indicates current bit number) for the first transition 0-1 or 1-0 (Step 1). When it is found, the number of "1" (or "0") is calculated in a range from bit 1 to bit $i$ + MBD (Step 2). This value corresponds to a partial result $R(t)$. Then, in Step 3, all data ($D$) bits in this selected range (bit 1 to bit $i$ + MBD) are set to "0" (or "1"). All of these three steps are repeated unless all required transitions ($t_{max}$) are found or all bits ($i_{max}$) are checked.

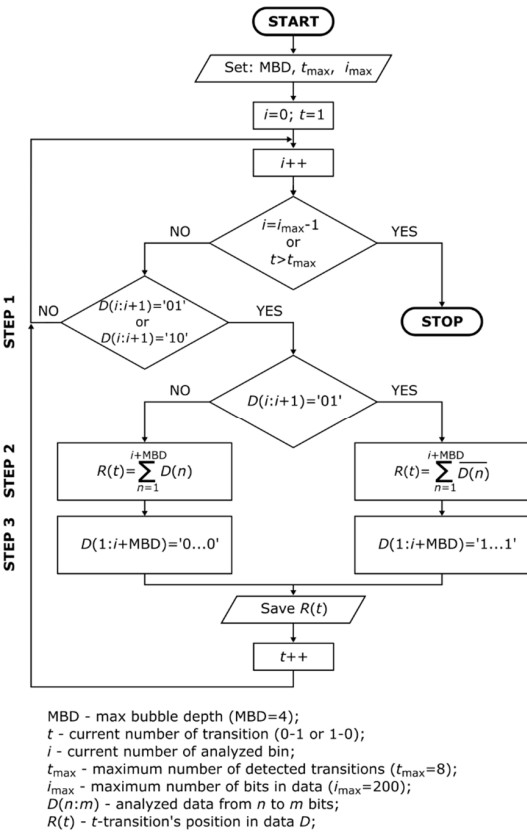

**Figure 4.** Counter-based pre-encoder algorithm: Step 1—finding transitions 0-1 or 1-0; Step 2—counting the number of logic states "0" or "1" in a selected range; Step 3—modification of the input data to search for another transition.

Let us analyze the practical example presented in Figure 3b. The first bit transition (0-1) is identified in bit 3. Then, we count number of "0s" in a range of data ($D$) from bit 1 to bit 3 + 5 = 8. There are five "0s" in this range so $R(1) = 5$. Next, all bits from 1 to 8 are set to "1" and above steps are repeated for transition 1-0.

This algorithm was first presented in [21] but it did not work properly due to a very high MBD and too short pulse widths in the wave union. The improved layout design (including manual placing and routing) allowed us to minimize the MBD to 5, which solved the problem. The presented algorithm allows for a relatively easy analysis of the obtained results but its hardware implementation requires many clock cycles to perform Steps 2 to 3 multiple times, depending on $t_{max}$. Thus, we propose another approach, better suited for TDC implementation in FPGA devices.

### 4.2. Pre-Encoder: Decomposition and Clustering-Based

Bubble errors can happen on the limited number of bits around the expected transition (defined as MBD). The decomposition method aims to spread these bits into different sub-sequences (sub-TDLs). Then, analyzing each sub-TDL separately, bubble errors are not visible anymore. Sub-TDLs are created by taking every $k$-bin from the binary code latched in a single register. Value $k$ must be at least equal to the MBD, and in our particular case it equals 5. The decomposition method principle is presented at the top of Figure 5.

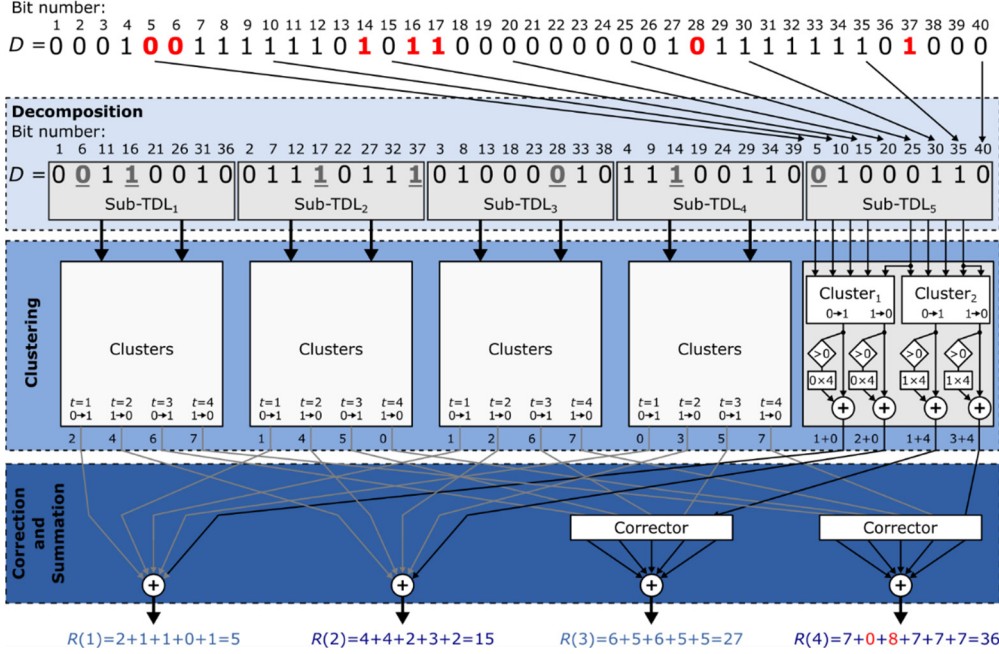

**Figure 5.** Pre-encoder with decomposition and clustering. Decomposition divides input data ($D$) to several sub-sequences (sub-TDLs) to eliminate bubble errors. Clustering is used to identify transitions occurrence in small chunks (four bits, $c = 4$) of the sub-TDLs. An additional (5th) cluster's input is used to identify boundary transitions (between clusters). Correction and summation merge results from all clusters. Correctors correct the result when the given transition was not detected in all clusters (e.g., value 8 is added to $R(4)$).

A cluster is a set of LUTs that can detect transitions in a certain number of bits ($c$). A cluster size (number of considered bits in a one cluster) is defined in such a way that there are no more than one transition 1-0 and one transition 0-1 in the analyzed part of the code. The cluster is able to find the position for each of these transitions. For example, a 4-input cluster transforms code "0110" to the following decimal results: 1 (for 0-1 transition) and 3 (for 1-0 transition). The clusters are multiplied to cover the sub-TDL range. An additional cluster's input is taken from the first input of a neighboring cluster. This allows finding transitions that occur on the border of the two clusters. Thus, the clusters size in

an example shown in Figure 5 is in fact 4 ($c = 4$), though in fact they have 5 inputs. In the last cluster, the last bit of the code is connected to this additional input. A single cluster gives results in a range from 0 to $c$, but the sub-TDL length can be even several times larger. Thus, non-zero results from clusters must be enlarged by adding a value $(p - 1) \times c$, where $p$ is a selected cluster number. All these operations are presented on the sample data in the middle of Figure 5.

Appropriate transition numbers ($t$) from clusters are added up to get $R(t)$. When the last transition occurs close to the end of a raw data code ($D$), then some sub-TDL can miss this transition. In Figure 5 such a case exists in a sub-TDL$_2$. To obtain the proper output result, a dedicated corrector is used to find whether a transition occurs in any of the clusters. If so, then 0 results from clusters are corrected by adding a value that equals to sub-TDLs size (8 in the example presented in Figure 5). Assuming that at least one transition 0-1 and one transition 1-0 are always latched, the corrector circuits are applied only for the third and subsequent transitions.

The pre-encoder based on decomposition and clustering can be implemented relatively easily in an FPGA device. The decomposition method requires only to reconnect signals between the TDC and the clusters. The clusters are implemented in the LUTs, a vital part of FPGA devices. In comparison to the encoder presented in the previous paragraph, this one can take advantage of parallelism and pipelining, which means that each step can be executed simultaneously for all bits and partial results can be stored in registers.

*4.3. Post-Encoder*

The pre-encoder allows detecting the position of each transition in code ($R(t)$). In the multisampling WU TDC, there are overall 4 registers and all of them can store multiple transitions. A post-encoder is used to convert all of this data into a single value.

After completing the measurement, Register 3 contains the first sample of the TDL state, Register 4 contains the second sample, Register 1 contains the third sample, and Register 2 the fourth one. Samples 1 and 2 always contain information about the first pulse in the wave union. When we add all $R(t)$s from these registers then it turns out that the higher the result, the longer the measured time interval. In Samples 3 and 4, the first pulse is not visible because it passed through the TDL before its state was sampled by the second pulse from the type A wave union launcher. This is shown in the example in Figure 6. Thus, the sum of $R(t)$s does not give clear information about the measured time interval. The role of the post-encoder is to correct results from registers 1 and 2 (Samples 3 and 4, respectively) according to related data stored in Registers 3 and 4 (Samples 1 and 2, respectively) and merge all results to a single output code.

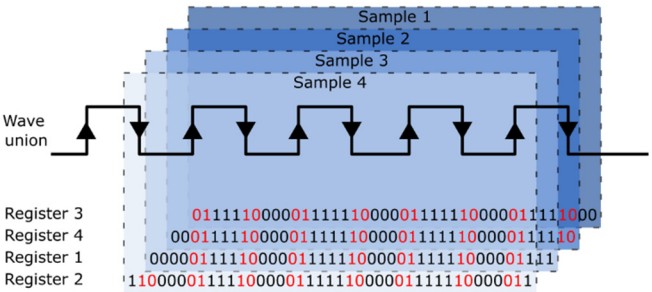

**Figure 6.** MSWU TDC operation principle. Register 1 (Sample 3) contains different number of transitions than other registers resulting in lower value of $\Sigma R_{\mathrm{samp3}}(t)$.

We executed 100,000 measurements of the calibration signal, then performed pre-encoding and finally add all transition positions for each sample stored in all registers: $\Sigma R_{\mathrm{samp1}}(t)$, $\Sigma R_{\mathrm{samp2}}(t)$, $\Sigma R_{\mathrm{samp3}}(t)$, $\Sigma R_{\mathrm{samp4}}(t)$. In terms of both Samples 1 and 2, the higher the $\Sigma R(t)$ value, the longer the time interval. Thus, we can add $\Sigma R_{\mathrm{samp1}}(t)$ to $\Sigma R_{\mathrm{samp2}}(t)$ and reorder obtained values from the smallest (shortest time interval) to the

highest (longest time interval—TDC operation range). Results latched in all registers for single measurements are strictly related to each other. Thus, we can also reorder $\Sigma R_{samp3}(t)$ and $\Sigma R_{samp4}(t)$ according to the sum: $\Sigma R_{samp1}(t)$ plus $\Sigma R_{samp2}(t)$. The results of this operation are presented in Figure 7a.

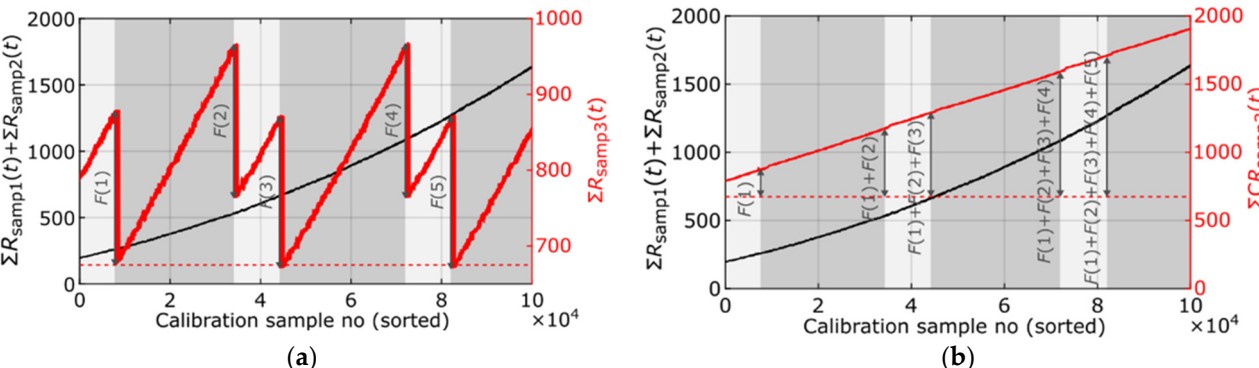

**Figure 7.** Post-encoder data processing: (**a**) before and (**b**) after correction. Correction factors (*F*) are added to selected $\Sigma R_{samp3}(t)$ and $\Sigma R_{samp4}(t)$ in order to obtain a monotonic transfer function.

Values $\Sigma R_{samp3}(t)$ and $\Sigma R_{samp4}(t)$ change non-monotonically with respect to increasing time interval duration. This is because Samples 3 and 4 can be taken when the beginning of the wave union (first pulse) already propagates through the TDL. In particular, for a one time interval there can be 8 transitions in the register, while for a bit longer time interval we can obtain 7 transitions because one pulse already flushed out from the TDL, while a new one has not appeared yet (see Figure 6). Hence the non-monotonicity in $\Sigma R_{samp3}(t)$ and $\Sigma R_{samp4}(t)$. The post-encoder aims to correct these values using information from $\Sigma R_{samp1}(t)$ plus $\Sigma R_{samp2}(t)$, to achieve monotonicity. The post-encoder first finds places of discontinuity in the sorted data with $\Sigma R_{samp3}(t)$ and $\Sigma R_{samp4}(t)$, and then adds correction factors (*F*) for selected ranges.

Correction factors are defined as differences between maximum and minimum values of $\Sigma R_{samp3}(t)$ (or $\Sigma R_{samp4}(t)$ in neighboring continuous ranges (calibration sample numbers, marked in grey in Figure 7a). After making corrections values of $\Sigma CR_{samp3}(t)$ and $\Sigma CR$samp4(*t*), where *CR* means results corrected by *F*, they become monotonic (Figure 7b). Final output codes (*OC*) for each calibration sample can then be calculated as:

$$OC = \sum_t R_{samp1}(t) + \sum_t R_{samp2}(t) + \sum_t CR_{samp3}(t) + \sum_t CR_{samp4}(t). \tag{1}$$

## 5. Measurement Results

In Section 4 we described how to obtain a single, decimal value from the MSWU TDC. Next, this result has to be translated to time domain. In the designed TDC we tested three cases. First, we investigated only the first transition in Register 1 ($R_{samp1}(1)$). This corresponds to the simple and popular digital method known as the flash TDC or the time coding line (TCL) [23]. In the second case, we investigated results only from Register 3 (sample 1: $\Sigma R_{samp1}(t)$). To some extent, we can regard this as a typical WU TDC (in fact, we have a different number of transitions depending on the measured time interval). The last case refers to the results obtained when all samples (results from all four registers) are taken into account (Equation (1), MSWU TDC). Note that for cases 1 and 2 we used the described pre-encoding procedure and for the last case we additionally employ the post-encoding too.

According to the principles of the SCDT, the number of occurrences of each of the codes during calibration can be divided by the calibration sample size and multiplied by the TDC operation range to determine the size of a given quantization step. The results of such calculation for all three mentioned cases are presented in Figure 8. For the MSWU TDC we obtained overall 3688 possible output codes. Taking into account the 1.43 ns TDC

operation range, in the configuration presented in Figure 2, the mean resolution equals 0.39 ps. Overall, the mean resolution is improved more than 25 times comparing to the TCL (from about 10.35 ps to 0.39 ps).

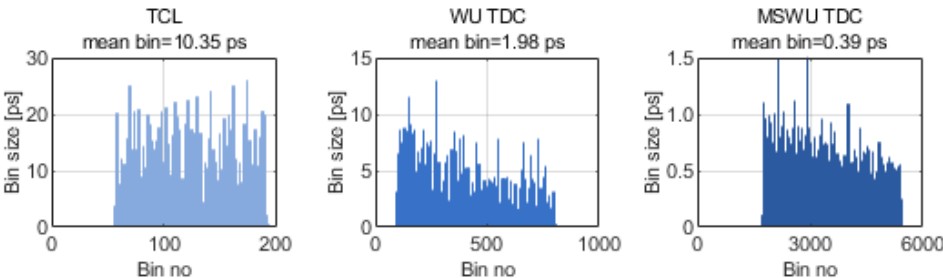

**Figure 8.** Quantization bin sizes for different TDCs.

Calibration data allow us to (1) construct the TDC transfer function and (2) define correction factors (*F*) as well as continuous ranges of $\Sigma R_{\mathrm{samp3}}(t)$ and $\Sigma R_{\mathrm{samp4}}(t)$. The transfer function is a cumulative sum of all bin sizes and is presented in Figure 9 for all three considered cases. The identified correction factors and continuous ranges are used to correct raw measurement data obtained in Register 1 and Register 2 (Samples 3 and 4, respectively) during measurements.

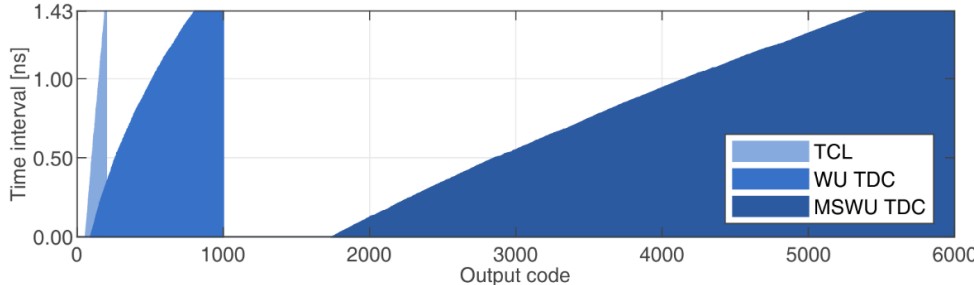

**Figure 9.** Transfer functions of different TDCs.

After completing the calibration, we executed 2000 measurements (1000 in each channel) of a constant time interval (approximately 0 s). A time interval between hit and the nearest edge of clock signal measured in a single TDC was calculated using equation:

$$\Delta t = \frac{b(OC)}{2} + \sum_{j=1}^{OC-1} b(j), \tag{2}$$

where *b(j)* is a *j*-th bin size and *OC* indicates obtained output code.

The value of the time interval between the pulses obtained by splitting a single pulse into two channels was calculated as the difference $\Delta t$ measured for the MSWU TDCs start and stop (see Figure 2). The obtained results are presented in Figure 10. Note that this time interval $\Delta t$ is in fact the difference between two measurements. Assuming that both implemented TDCs have similar performance we can estimate a single shot precision (SSP) as a standard deviation of obtained results divided by a square root of 2.

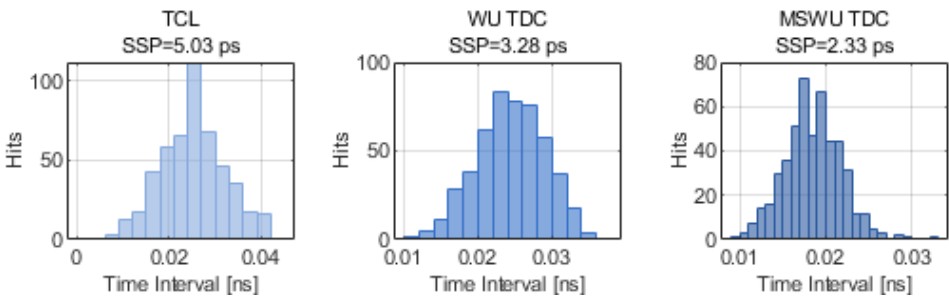

**Figure 10.** Single-shot precision test for different TDCs using a signal split into two channels.

Using the simple and popular TCL method we achieved SSP of about 5 ps in the test setup presented above, however, using the newly designed MSWU TDC improves the SSP more than twice (to 2.33 ps). When compared to results presented in [21] we did not observe any outliers. This proves that the presented algorithm works efficiently. Furthermore, the same results were obtained using both described pre-encoders with one common post-encoder.

In the last test we have used two outputs of the time interval generator BNC Model 745 to produce different time intervals within the MSWU TDC operation range (1.43 ps). Obtained results are presented in Figure 11. The presented SSP is slightly worse than the one presented in Figure 10 due to a timing jitter between start and stop pulses produced by the source generator. However, the presented algorithm still allows us to maintain better than 2.9 ps SSP obtained in a mid-range FPGA device (Xilinx Kintex-7).

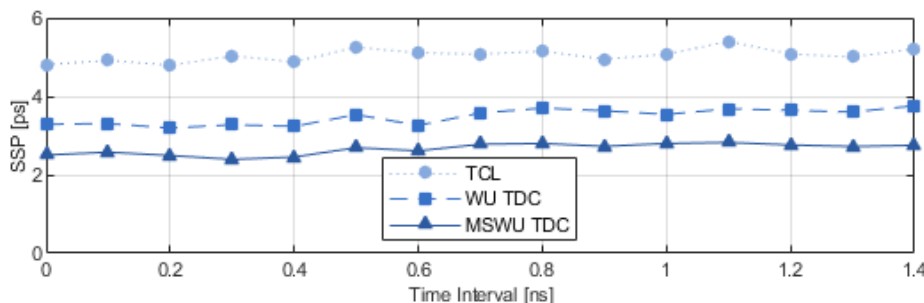

**Figure 11.** Single-shot precision test for different TDCs and for various time intervals.

## 6. Discussions

The very high SSP (2.33 ps) in the TDC was obtained in a very specific test setup in which all potential sources of additional errors were minimized. This confirms both the proper operation of the developed encoding algorithm and the great measurement potential of the MSWU TDC. Comparing to other related works, we have shown how to take advantage of all transitions that can occur in the raw data from the WU TDC, rather than skipping them. This is vital for the presented MSWU TDC but can also improve the performance of the WU TDC based on the infinite step response launchers.

So far, all the calculations presented have been performed in MATLAB and require the transmission of 800-bits of data for each single measurement (4 registers, 200 flip-flops each). The next step is to move the presented algorithm directly to hardware. For this purpose, the pre-encoder with decomposition and clustering functions was developed. The decomposition method changes the order of outputs from the registers. It does not require any logic elements such as LUTs or flip-flops, only interconnection resources. Clusters are made as bunches of LUTs connected to clustering stage adders (see Figure 5). A single 4-bit cluster ($c = 4$, as in Figure 5) requires three 5-input LUTs (result 0—no transition, results 1-4 –positions of transitions). Thus, implementation of four 200-bits registers requires $4 \times 50 \times 3 = 600$ LUTs. The most time-demanding part of encoding algorithm is that related to the operation of multibit adders that merge results from the clusters. However, it

is already proven that using multiple pipeline stages these circuits can operate with very high throughput, e.g., 277 MSa/s [13]. Optimal cluster size as well as a number of pipeline stages will be the subject of further considerations during hardware implementation of the pre-encoder to find trade-off between logic resource utilization and throughput. The hardware pre-encoder, solving bubble errors and allowing to detect variable number of transitions, can be directly used in most WU TDCs as well as in the TCL-based TDCs (see Figures 8–11). The post-encoder can further boost TDC performance in solutions based on WU type B and MSWU.

In contrast to the pre-encoder, the post encoder is not easily transferable to the hardware. However, for the presented TDC design, the pre-encoder allows minimizing the amount of data that needs to be transferred more than 18-times, from $4 \times 200$-bits to $4 \times 11$-bits (a sum of 8 transition positions where 8 bits are required to store a single position in a data composed of 200-bits). The computational cost of this part is strongly related to the number of calibration samples. During our experiment we have applied 100,000 measurement samples to build the transfer function (sorting results from registers and adding correction factors *F*). This calculation, performed without any optimization and after storing all required samples, took several seconds in MATLAB. In addition, described procedure must be executed only once. Then, during measurements, the post-encoder operates much faster, only adding correction factors and then reading final time interval values from the stored transfer function.

### 7. Conclusions

An efficient hardware encoder is still of interest to researchers working on the WU TDCs. This is especially important taking into account new emerging architectures such as the MSWU TDC. The presented 2-step software algorithm, with pre- and post-encoders, is designed to be at least partially implemented in hardware. Its tests confirm the algorithm's proper and error-free operation. The algorithm can be used in any kind of the WU TDCs that are already applicable in LIDAR systems, measurement instrumentations, or nuclear physics. The pre-encoder with decomposition and clustering can be directly applied to WU TDCs based on launchers with finite step response (type A). Combined with the post-encoder it can enhance also the performance of WU TDCs based on launchers with infinite step response (type B). This will be achieved by taking into account all valid transitions in the output code, not just one or two as it was presented in previous works. Additionally, the algorithm allows us to take full advantage of the newly developed MSWU TDC by obtaining one of the highest performance in such kind of converters implemented in FPGA devices (single shot precision better than 2.9 ps, while the resolution as high as 0.39 ps). Although this paper is exclusively focused on data encoding problem, in future work we are going to transfer the pre-encoder to hardware and provide a full characterization of the newly designed MSWU TDC.

**Author Contributions:** Conceptualization, P.K.; methodology, P.K.; software, P.K. and D.S.; validation, P.K.; formal analysis, P.K., D.S. and R.S.; investigation, P.K., D.S. and R.S; resources, R.S.; data curation, P.K. and D.S.; writing—original draft preparation, P.K.; writing—review and editing, P.K., D.S. and R.S.; visualization, P.K. and D.S.; supervision, R.S.; project administration, P.K. and R.S.; funding acquisition, R.S. All authors have read and agreed to the published version of the manuscript.

**Funding:** This work was financed by the Military University of Technology under research project UGB no 851/2021 entitled "Adaptive calibration of high-precision integrated time counters".

**Conflicts of Interest:** The authors declare no conflict of interest.

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
