# Peer review of "Bubble-Proof Algorithm for Wave Union TDCs"

_electronics, doi:10.3390/electronics11010030_

Round 1

Reviewer 1 Report

An effective algorithm used to decode raw output data from a newly designed multisampling wave union time to digital converter (TDC) is described in the paper. This algorithm is able to correct bubble errors and detect any number of transitions which occur in the wave union TDC output code. The algorithm has the advantage that it can be used for any kind of wave union TDCs and is intended for partial hardware implementation.

The paper topic is interesting, and the manuscript is comprehensive. The paper is well documented, and bibliographic references are comprehensive. The results seem coherent and described with sufficient clarity. The main contributions of the paper are clearly highlighted. It is noted that there are real perspectives for research development.

Consequently, the paper may be recommended for publication in Electronics. But, before publication, I recommend the revision of the conclusions section. These are dry, without highlighting the significant contributions of the paper. Also, the text and English language in the paper must checked, edited, and corrected.

Author Response

Response to Reviewer 1

Thank you for your review of our work and your overall positive feedback. Below you can find our response to each of the comments and information about the actions taken.

REMARK 1

Consequently, the paper may be recommended for publication in Electronics. But, before publication, I recommend the revision of the conclusions section. These are dry, without highlighting the significant contributions of the paper. Also, the text and English language in the paper must checked, edited, and corrected.

RESPONSE: Thank you for valuable remark, we have rewritten conclusions.

ACTION: Section 7 is changed as follows:

An efficient hardware encoder is still of interest to researchers working on the WU TDCs. This is especially important taking into account new emerging architectures such as the MSWU TDC. The presented 2-step software algorithm, with pre- and post-encoders, is designed to be at least partially implemented in hardware. Its tests confirm the algorithm’s proper and error-free operation. The algorithm can be used in any kind of the WU TDCs that are already applicable in LIDAR systems, measurement instrumentations or nuclear physics. The pre-encoder with decomposition and clustering can be directly applied to WU TDCs based on launchers with finite step response (type A). Combined with the post-encoder it can enhance also the performance of WU TDCs based on launchers with infinite step response (type B). This will be achieved by taking into account all valid transitions in the output code, not just one or two as it was presented in previous works. Additionally, the algorithm allows us to take full advantage of the newly developed MSWU TDC by obtaining one of the highest performance in such kind of converters implemented in FPGA devices (single shot precision better than 2.9 ps, while the resolution as high as 0. 39 ps). Although this paper is exclusively focused on data encoding problem, in future work we are going to transfer the pre-encoder to hardware and provide a full characterization of the newly designed MSWU TDC.

REMARK 2

Also, the text and English language in the paper must checked, edited, and corrected.

ACTION: The article has been double-checked for linguistic correctness.

Reviewer 2 Report

The authors focus their study on the time to digital converters which are mainly used in the field programmable gate arrays devices and they use different variations of the wave union method.

Specifically, the authors describe in detail an effective algorithm in order to decode raw output data from a newly designed multi sampling wave union time to digital converter.

The provided discussion and analysis is thorough and the authors have well thought out their main contributions.

The structure of the paper is good in order to enable the reader to easily follow it. The title of the paper is matching very well the description of the theoretical analysis that is presented.

The authors have presented some indicative numerical results in order to show the pure operation and the performance of the proposed algorithm.

The authors should consider the following suggestions provided by the reviewer in order to improve the scientific depth of their manuscript, as well as the quality of its presentation.

Initially, the authors should discuss some realistic applications where the proposed time to digital converters embedded in the FPGAs can find applications, such as Fragkos, G., et al. "Artificially Intelligent Electronic Money." IEEE Consumer Electronics Magazine 10.4 (2020): 81-89, in order to show the importance of the proposed research.

Furthermore, the authors should include an additional subsection in their manuscript providing the theoretical analysis of the computational complexity of the proposed framework as well as discussing the corresponding implementation cost.

Finally, the overall manuscript should be checked for typos, syntax, and grammar errors in order to improve the quality of its presentation.

Author Response

Response to Reviewer 2

Thank you for your review of our manuscript and your overall positive feedback. Below you can find our response to each of the comments and information about the actions taken.

REMARK 1

Initially, the authors should discuss some realistic applications where the proposed time to digital converters embedded in the FPGAs can find applications, such as Fragkos, G., et al. "Artificially Intelligent Electronic Money." IEEE Consumer Electronics Magazine 10.4 (2020): 81-89, in order to show the importance of the proposed research.

RESPONSE: In the original manuscript we have given few examples of application where wave union time-to-digital converters are used: This method was first proposed by Wu and Shi in 2008 [1], and since then it has been used for various applications including nuclear physics [2], [3], time interval counters [4], light detection and ranging (LiDAR) systems [5].

We have carefully analyzed the proposed publication. It is really interesting, but unfortunately, it does not fit well as an example application to our converter. Precise time measurement is indeed very important in financial services to prevent abuse. However, until now, synchronization in such services has been performed with much lower accuracy (i.e. ns-level) than provided by our converter (ps-level). We believe that the converter can be rather used for nuclear experiments e.g. in CERN or FermiLab, or as a part of sophisticated measurement instrumentations (precise time interval counters, time analyzers, frequency counters, etc.).

ACTION: In Conclusions, We have additionally stressed-out information about potential applications:

The algorithm can be used in any kind of the WU TDCs that are already applicable in LIDAR systems, measurement instrumentations or nuclear physics.

REMARK 2

Furthermore, the authors should include an additional subsection in their manuscript providing the theoretical analysis of the computational complexity of the proposed framework as well as discussing the corresponding implementation cost.

RESPONSE: So far we have focused on developing algorithms and we have not made any optimization. Now our goal is to implement the pre-encoder in hardware. The computational complexity as well as the corresponding implementation cost will be very important then. During implementation we can make some trade-off between logical resource utilization and measurement rate. For example, we can add pipeline stages to the encoder to improve throughput at the cost of latency and logical resource occupation. At current stage we can only give some rough estimation about the algorithm performance.

ACTION: Section 6 is extended as follows:

So far, all the calculations presented have been performed in MATLAB and require the transmission of 800-bits of data for each single measurement (4 registers, 200 flip-flops each). The next step is to move the presented algorithm directly to hardware. For this pur-pose, the pre-encoder with decomposition and clustering functions was developed. The decomposition method changes the order of outputs from the registers. It does not require any logic elements such as LUTs or flip-flops, only interconnection resources. Clusters are made as bunches of LUTs connected to clustering stage adders (see Figure 5). A single 4-bit cluster (c=4, as in Figure 5) requires three 5-input LUTs (result 0 – no transition, results 1-4 –positions of transitions). Thus, implementation of four 200-bits registers requires 4×50×3=600 LUTs. The most time-demanding part of encoding algorithm is that related to the operation of multibit adders that merge results from the clusters. However, it is already proven that using multiple pipeline stages these circuits can operate with very high throughput, e.g. 277 MSa/s [13]. Optimal cluster size as well as a number of pipeline stages will be the subject of further considerations during hardware implementation of the pre-encoder to find trade-off between logic resource utilization and throughput. The hardware pre-encoder, solving bubble errors and allowing to detect variable number of transitions, can be directly used in most WU TDCs as well as in the TCL-based TDCs (see Figures 8-11). The post-encoder can further boost TDC performance in solutions based on WU type B and MSWU.

In contrast to the pre-encoder, the post encoder is not easily transferable to the hardware. However, for the presented TDC design, the pre-encoder allows minimizing the amount of data that needs to be transferred more than 18-times, from 4 × 200-bits to 4 × 11 bits (a sum of 8 transition positions where 8 bits are required to store a single position in a data composed of 200 bits). The computational cost of this part is strongly related to the number of calibration samples. During our experiment we have applied 100 000 measurement samples to build the transfer function (sorting results from registers and adding correction factors F). This calculation, performed without any optimization and after storing all required samples, took several seconds in MATLAB. In addition, described procedure must be executed only once. Then, during measurements, the post-encoder operates much faster, only adding correction factors and then reading final time interval values from the stored transfer function.

REMARK 3

Finally, the overall manuscript should be checked for typos, syntax, and grammar errors in order to improve the quality of its presentation.

ACTION: The article has been once more carefully revised in search of typos, syntax, and grammar errors.

Round 2

Reviewer 2 Report

The authors have addressed in detail the reviewers’ comments and the quality of presentation, as well as the scientific depth of the paper have been substantially improved. This reviewer has no further concerns regarding the publication of this paper.